# Correlations between Environmental Factors and the Distribution of Juvenile *Hucho bleekeri* in the Taibai River, Shaanxi, China

Jinming Wu [1], Jinping Wu [1], Huan Ye [1], Wei Xiong [1], Wanmin Qu [1,2], Xiaoqian Leng [1,*] and Hao Du [1]

[1] Key Laboratory of Freshwater Biodiversity Conservation, Ministry of Agriculture and Rural Affairs of China, Yangtze River Fisheries Research Institute, Chinese Academy of Fishery Sciences, Wuhan 430223, China; jinming@yfi.ac.cn (J.W.); wujinping@yfi.ac.cn (J.W.); yehuan@yfi.ac.cn (H.Y.); taihuxiongwei@163.com (W.X.); 18674049710@163.com (W.Q.); duhao@yfi.ac.cn (H.D.)

[2] Fisheries College, Huazhong Agricultural University, Wuhan 430070, China

* Correspondence: lengxiaoqian@yfi.ac.cn

**Abstract:** Sichuan taimen (*Hucho bleekeri*) is a national highly protected wild animal that faces significant impacts from habitat degradation and loss. To study the habitat selection by *H. bleekeri*, data on the Taibai River and the distribution of juvenile *H. bleekeri* were recorded seasonally between February 2017 and January 2018, and habitat selection was assessed using the suitability curve method. The results indicate that the average distribution density of juvenile *H. bleekeri* in the Taibai River is 0.08 ± 0.09 ind./m, with an optimal elevation range from 1200 to 1600 m, river sinuosity range from 1.0 to 1.2, and vegetation coverage between 0.7 and 1.0. When choosing a microhabitat, juvenile *H. bleekeri* individuals tend to inhabit water areas with a water depth of 0.65 ± 0.33 m, flow velocity of 0.50 ± 0.24 m/s, and offshore distance of 7.66 ± 4.25 m. Furthermore, smaller juvenile fish prefer nearshore habitats with lower flow velocities and shallower water depths. The results provide technical support for the protection and restoration of the habitat of *H. bleekeri*.

**Keywords:** resource density; habitat characteristics; correlation analysis

**Key Contribution:** In this study, the distribution and habitat requirements of juvenile *H. bleekeri* in the Taibai River were determined using a literature review and field investigation.





## 1. Introduction

The protection of endangered species is a popular topic in biodiversity research [1]. Habitat degradation caused by multiple human impacts and illegal fishing are the main factors of risk [2]. Therefore, implementing measures to contrast illegal fishing and protecting habitats are key priorities for protecting endangered fish species [3]. Compared with the prohibition of fishing, research required for habitat protection is much more complex. Understanding the habitat requirements and environmental selection preferences of endangered fish is the basis for formulating habitat protection plans. Habitat characteristics and environmental factors determine the species distribution and population abundance [4,5]. As fish habitats, aquatic ecosystems have environmental characteristics, such as river morphology, water temperature, water velocity, and substrate type, that are closely related to the survival and reproduction of fish [6–8].

The Sichuan taimen (*Hucho bleekeri*), belonging to Salmoniformes, Salmonidae, is a typical landlocked species that inhabits the edges of clear-water rivers at high altitudes (≥1000 m) with low water temperatures and high oxygen content. Its speciation and distribution characteristics play an important role in studying zoogeography, paleoecology, and relationship between evolutionary systems of fish and climate change [9]. Historically, *H. bleekeri* has been mainly distributed in Sichuan (upper reaches of the Dadu and Min

rivers), Qinghai (Make River), and Shaanxi (Taibai and Xushui rivers and upper tributaries of the Han River) [10,11]. Since the 1960s, the population has sharply decreased because of climate change and human activity. To date, there have been few records of *H. bleekeri* in most historical areas [12,13]. Due to its endangered status, *H. bleekeri* has been listed as a critically endangered species on the International Union for Conservation of Nature Red List and as a national first-class protected aquatic wildlife in 2021 [14]. Fortunately, several *H. bleekeri* wild populations were identified in the Taibai River in 2012 [15]. Therefore, there is an urgent need for the protection of and research on the natural resources used by *H. bleekeri*.

Limited by the endangered status of this species, research on the natural population of *H. bleekeri* was mainly carried out in the 1990s, including research into their biology, distribution area, habitat characteristics, and resource identification [16–18]. Similar to other salmon species, *H. bleekeri* tend to inhabit mountain streams with low water temperatures, clear water quality, and dense vegetation [18]. However, there are differences in the selection of specific environmental parameters among species or different life history stages. Adult fish generally exhibit relatively broad environmental adaptability, while their environmental selectivity is stronger during the breeding and juvenile stages [19]. During the breeding period, shallow, sandy, and slightly flowing river sections upstream are generally chosen as spawning grounds [20]. During the juvenile stage, fish prefer river sections with abundant food resources, offering sheltered structures to forage for food, and predator avoidance [21]. Therefore, we hypothesized that the abundance of juvenile *H. bleekeri* should be highly correlated with several environmental variables.

The Taibai River is a fourth-level tributary of the Yangtze River that flows into the Hongyan, Bao, and Han rivers. It has a total length of 59.4 km and a drainage area of 377.87 km$^2$. The average gradient is 30.9‰, and the forest coverage rate reaches 89.5%. To further understand the habitat requirements of *H. bleekeri*, the distribution characteristics and habitat preferences of these existing juvenile populations were investigated at the river, reach, and microhabitat scales; correlations between the occurrence of juvenile *H. bleekeri* and environmental factors were analyzed; and environmental factors affecting the distribution of juvenile *H. bleekeri* in the Taibai River were determined. The results of this study provide guidance for habitat conservation and habitat amendment and contribute to population restoration.

## 2. Materials and Methods

### 2.1. Watershed Environmental Survey

A survey was conducted on the Taibai River between February 2017 and January 2018. The water temperature was monitored daily using an automatic recorder. Seasonal surveys were conducted on other physical and chemical variables of watershed waterbodies, including the dissolved oxygen, flow velocity, and transparency, using a portable water-quality monitor (YSI, Yellow Springs, OH, USA), and also, aquatic biological resources such as fish, zooplankton, phytoplankton, and benthic animals were assessed using methods proposed by Meng et al. [22]. Two survey sections were established upstream and downstream in the river, respectively.

### 2.2. Reach Characteristics and Juvenile Fish Distribution

Thirty-five sampling sites were established in the upstream and downstream areas. The distribution of the sampling sites is shown in Figure 1.

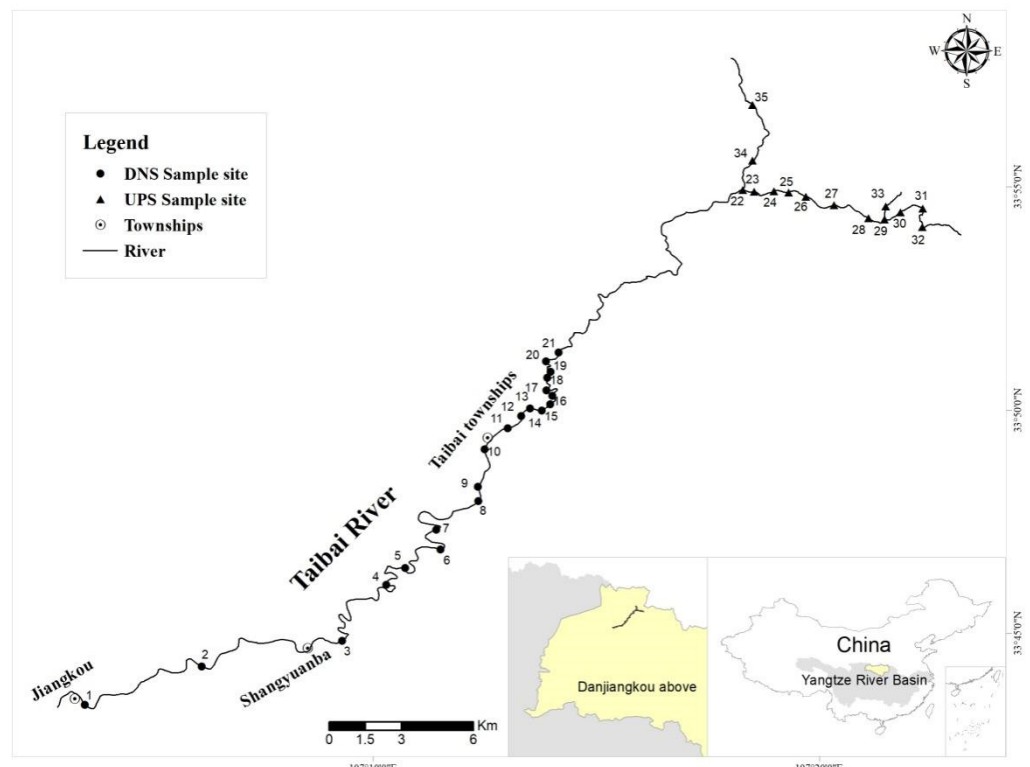

**Figure 1.** Distribution of sampling sites in Taibai River.

### 2.2.1. Juvenile Fish Collection

Juvenile *H. bleekeri* were collected via electrofishing in October 2017. The specific methods were as follows: A 50 m long survey reach was set for each sampling site, and a net with a mesh size of 2 cm was set up in the upstream and downstream of the survey reach to prevent fish from escaping. A knapsack electrofishing machine (4500 W) was used to fish from downstream to upstream with two passes. The species in the caught samples were identified and the number of juvenile *H. bleekeri* (<20 cm) was recorded immediately. Subsequently, the samples were returned to the river. Finally, the density of juvenile *H. bleekeri* was calculated as the number of individuals per meter of stream distance.

### 2.2.2. Measurement of Habitat Variables

After fish sampling, habitat variables, including elevation, river sinuosity, river wetting rate, vegetation coverage, and river habitat diversity, were estimated at each sampling site. The measurement and calculation methods for each variable are as follows: The elevation was measured using a differential GPS (NavCom, Torrance, CA, USA). The river wetting rate is the ratio of the water surface width to the full width of the river. The water surface width was measured using a leather measuring tape. The full-water width was determined based on the river extension position and flood hydrographs. These variables were measured three times, at the starting point, midpoint, and endpoint of each site, and the mean value was determined. River sinuosity is the ratio of the actual length to the straight-line length of a reach. The actual length was measured in sections using a leather measuring tape, and the straight-line length was determined using a laser range finder. Vegetation coverage is the ratio of shrubs, grassland, and tree vegetation within a 50 m region on both banks of the river. Regular areas were measured using an acre-measuring instrument, and irregular sizes were estimated via visual inspection. Habitats can be classified as pools, glides, riffles, side channels, or valley floor tributaries [23]. River habitat

diversity (H) represents the type and frequency of the occurrence of various habitat units within a river [23]. The calculation equation is as follows:

$$H = T \times N, \tag{1}$$

where T is the number of habitat units and N is the number of habitat unit types.

The suitability curve was plotted based on juvenile density and measured environmental factors. All above-mentioned analyses were conducted using R software (R version 3.6.3).

### 2.3. Microhabitat Characteristics of Juvenile Fish
2.3.1. Underwater Shot

Sample sites with a dense distribution of juveniles were divided into grids of 1 m × 1 m. An underwater camera was installed at the four nodes of each grid, with an angle covering the entire grid area to record the movements of the juveniles. Juveniles were categorized into three groups according to the body length estimated from the length of the background substrate: 5 to 10, 11 to 15, and 16 to 20 cm. After the observation, the grid was marked with colored stones at the center and boundary of the juvenile habitat.

2.3.2. Measurement of Microhabitat Variables

The microhabitat variables, that is, substrate type, velocity, water depth, illuminance, and offshore distance, scored at 50 marked distribution sites were then analyzed according to video data and situ measurement. The size of the substrate in which juvenile feeding was measured by tape and the type was differentiated according to Cummins' method [24]. They were divided into gravel (2 to 16 mm, Gr), pebble (16 to 64 mm, Pe), cobble (64 to 256 mm, Co), boulders (>256 mm, Bo), and bedrock (Br). Water depth is the average water depth of the habitat center and the activity range of the juveniles. Illuminance was measured using an underwater illuminometer (ZDS-10). Offshore distance is the distance from the center of the juvenile habitat to the nearest bank.

Statistical analysis and data plotting were completed using SPSS software (version 26.0; SPSS, Inc., Chicago, IL, USA), and the nominal Type I error rate was set at 0.05.

### 3. Results
### 3.1. Environmental Characteristics of the Taibai River

The results of this study showed that the annual mean water temperature of the Taibai River was 11.6 ± 5.6 °C, with a range from 0.1 to 22.5 °C. The annual average dissolved oxygen content was 8.76 ± 0.46 mg/L (mean ± SD), with a range from 7.52 to 9.43 mg/L. The average flow velocity was 1.69 ± 0.41 m/s, with a range from 0.88 to 2.45 m/s. The water transparency was high, reaching more than 1.5 m in deep pools.

A survey of aquatic biological resources revealed that the composition of fish species in the Taibai River was relatively simple, with the largest populations being *Phoxinus lagowskii* and *Triplophysa* sp., accounting for 80.52% of the total catch. Zooplankton included protozoa, copepods, rotifers, and cladocerans. Phytoplankton was dominated by diatoms, accounting for 70.21% of the total species. Insects displayed the highest number of benthic animals, accounting for 82.14% of the total species (Table 1). The biomass and species composition of the above feed organisms were relatively high (Table 2).

**Table 1.** Composition of aquatic biological resources.

|  | Species | Individual Number | Species Number | Proportion (%) |
|---|---|---|---|---|
| Fish | *Phoxinus lagowskii* | 380 | | 51.77 |
| | *Triplophysa* sp. | 211 | | 28.75 |
| | *Brachymystax lenok tsinlingensis* | 58 | | 7.9 |
| | *Cobitis sinensis* | 49 | | 6.68 |
| | *Hucho bleekeri* | 36 | | 4.9 |
| Zooplankton | Protozoan | | 7 | 53.85 |
| | Copepoda | | 3 | 23.08 |
| | Rotifer | | 2 | 15.38 |
| | Cladocera | | 1 | 7.69 |
| Phytoplankton | Bacillariophyta | | 33 | 70.21 |
| | Chlorophyta | | 8 | 17.02 |
| | Cyanophyta | | 2 | 4.26 |
| | Chrysophyta | | 2 | 4.26 |
| | Xanthophyta | | 1 | 2.13 |
| | Euglenophyta | | 1 | 2.13 |
| Benthic animals | Oligochaeta | | 1 | 3.57 |
| | Gastropoda | | 2 | 7.14 |
| | Insecta | | 23 | 82.14 |
| | Turbellaria | | 1 | 3.57 |
| | Nematoda | | 1 | 3.57 |

**Table 2.** Seasonal statistics of the biomass of food organisms.

|  | Spring | Summer | Autumn | Winter | Average |
|---|---|---|---|---|---|
| Zooplankton (mg/L) | 0.17 ± 0.09 | 0.21 ± 0.08 | 0.16 ± 0.09 | 0.25 ± 0.08 | 0.20 ± 0.09 |
| Phytoplankton (mg/L) | 1.37 ± 0.37 | 2.34 ± 0.52 | 1.62 ± 0.23 | 0.89 ± 0.16 | 1.55 ± 0.63 |
| Benthic animals (g/m$^2$) | 2.88 ± 1.33 | 2.47 ± 1.51 | 2.83 ± 1.07 | 2.83 ± 1.28 | 2.75 ± 1.32 |

All data were presented as mean ± SD.

*3.2. Correlation between the Juvenile H. bleekeri Distribution and Environmental Characteristics in the Reach*

In total, 59 juvenile *H. bleekeri* individuals were collected for the survey (Table 3). According to the sample estimation method, the total number of juvenile *H. bleekeri* in the Taibai River was estimated to be 2528. The density ranged from 0 to 0.30 ind./m at various sampling sites, and the average distribution density was 0.08 ± 0.09 ind./m. The average density in the upstream and downstream was 0.08 ± 0.08 and 0.09 ± 0.10 ind./m, respectively. There were no significant differences in the distribution densities between the two sections.

The environmental factors at various points showed that the elevation of the survey area ranged from 920 to 1909 m, with an average of 1395 ± 278 m. The river sinuosity varied from 1.02 to 1.32, with an average of 1.13 ± 0.08. The river wetting rate ranged from 0.35 to 0.95, with an average of 0.80 ± 0.16. The vegetation coverage varied from 0.22 to 0.98, with an average of 0.76 ± 0.20. The river habitat diversity ranged from 2 to 24, with an average of 11 ± 6.

**Table 3.** Juvenile density and value of environmental factors in different sampling sites.

| Area | Sampling Site | Juvenile Number | Juvenile Density (ind./m) | Elevation (m) | River Sinuosity | River Wetting Rate | Vegetation Coverage | River Habitat Diversity |
|---|---|---|---|---|---|---|---|---|
| Downstream | 1 | 0 | 0 | 920 | 1.03 | 0.35 | 0.22 | 4 |
| | 2 | 0 | 0 | 960 | 1.02 | 0.94 | 0.65 | 4 |
| | 3 | 0 | 0 | 1018 | 1.05 | 0.42 | 0.54 | 6 |
| | 4 | 0 | 0 | 1094 | 1.28 | 0.88 | 0.92 | 8 |
| | 5 | 0 | 0 | 1130 | 1.14 | 0.78 | 0.58 | 4 |
| | 6 | 0 | 0 | 1146 | 1.03 | 0.56 | 0.54 | 6 |
| | 7 | 0 | 0 | 1157 | 1.08 | 0.87 | 0.48 | 4 |
| | 8 | 1 | 0.05 | 1179 | 1.17 | 0.93 | 0.86 | 8 |
| | 9 | 0 | 0 | 1197 | 1.12 | 0.63 | 0.43 | 6 |
| | 10 | 2 | 0.1 | 1207 | 1.03 | 0.65 | 0.22 | 6 |
| | 11 | 3 | 0.15 | 1221 | 1.18 | 0.75 | 0.85 | 12 |
| | 12 | 1 | 0.05 | 1235 | 1.14 | 0.72 | 0.97 | 10 |
| | 13 | 0 | 0 | 1237 | 1.04 | 0.49 | 0.87 | 6 |
| | 14 | 0 | 0 | 1249 | 1.05 | 0.85 | 0.68 | 8 |
| | 15 | 2 | 0.1 | 1253 | 1.22 | 0.91 | 0.95 | 15 |
| | 16 | 6 | 0.3 | 1254 | 1.04 | 0.93 | 0.86 | 12 |
| | 17 | 5 | 0.25 | 1279 | 1.08 | 0.78 | 0.93 | 16 |
| | 18 | 5 | 0.25 | 1295 | 1.16 | 0.85 | 0.98 | 18 |
| | 19 | 3 | 0.15 | 1297 | 1.15 | 0.95 | 0.83 | 15 |
| | 20 | 4 | 0.2 | 1299 | 1.09 | 0.61 | 0.72 | 6 |
| | 21 | 4 | 0.2 | 1304 | 1.17 | 0.88 | 0.78 | 16 |
| Upstream | 22 | 2 | 0.1 | 1547 | 1.22 | 0.94 | 0.92 | 16 |
| | 23 | 1 | 0.05 | 1571 | 1.13 | 0.9 | 0.96 | 18 |
| | 24 | 5 | 0.25 | 1599 | 1.04 | 0.79 | 0.96 | 20 |
| | 25 | 1 | 0.05 | 1614 | 1.04 | 0.83 | 0.89 | 24 |
| | 26 | 2 | 0.1 | 1630 | 1.15 | 0.66 | 0.93 | 18 |
| | 27 | 5 | 0.25 | 1670 | 1.21 | 0.79 | 0.83 | 15 |
| | 28 | 3 | 0.15 | 1729 | 1.06 | 0.93 | 0.86 | 18 |
| | 29 | 0 | 0 | 1762 | 1.32 | 0.89 | 0.87 | 20 |
| | 30 | 0 | 0 | 1816 | 1.11 | 0.9 | 0.95 | 12 |
| | 31 | 2 | 0.1 | 1868 | 1.32 | 0.83 | 0.92 | 12 |
| | 32 | 0 | 0 | 1909 | 1.24 | 0.93 | 0.77 | 4 |
| | 33 | 2 | 0.1 | 1833 | 1.25 | 0.86 | 0.68 | 6 |
| | 34 | 0 | 0 | 1596 | 1.09 | 0.95 | 0.54 | 2 |
| | 35 | 0 | 0 | 1755 | 1.13 | 0.92 | 0.65 | 4 |
| Average | | | 0.08 ± 0.09 | 1395 ± 278 | 1.13 ± 0.08 | 0.80 ± 0.16 | 0.76 ± 0.20 | 11 ± 6 |

Some data were presented as mean ± SD.

Correlations between environmental factors and the juvenile density were analyzed and presented using a suitability curve. The results showed that juvenile *H. bleekeri* had a highest population density at an elevation range from 1200 to 1600 m, a sinuosity value ranging from 1.0 to 1.2, vegetation coverage ranging from 0.7 to 1.0, and river habitat diversity between 10 and 20, indicating that the juvenile distribution was closely related to the above environmental factors. The optimum range of river wetting rate for juvenile distribution was not identified, and there was no obvious correlation between them (Figure 2).

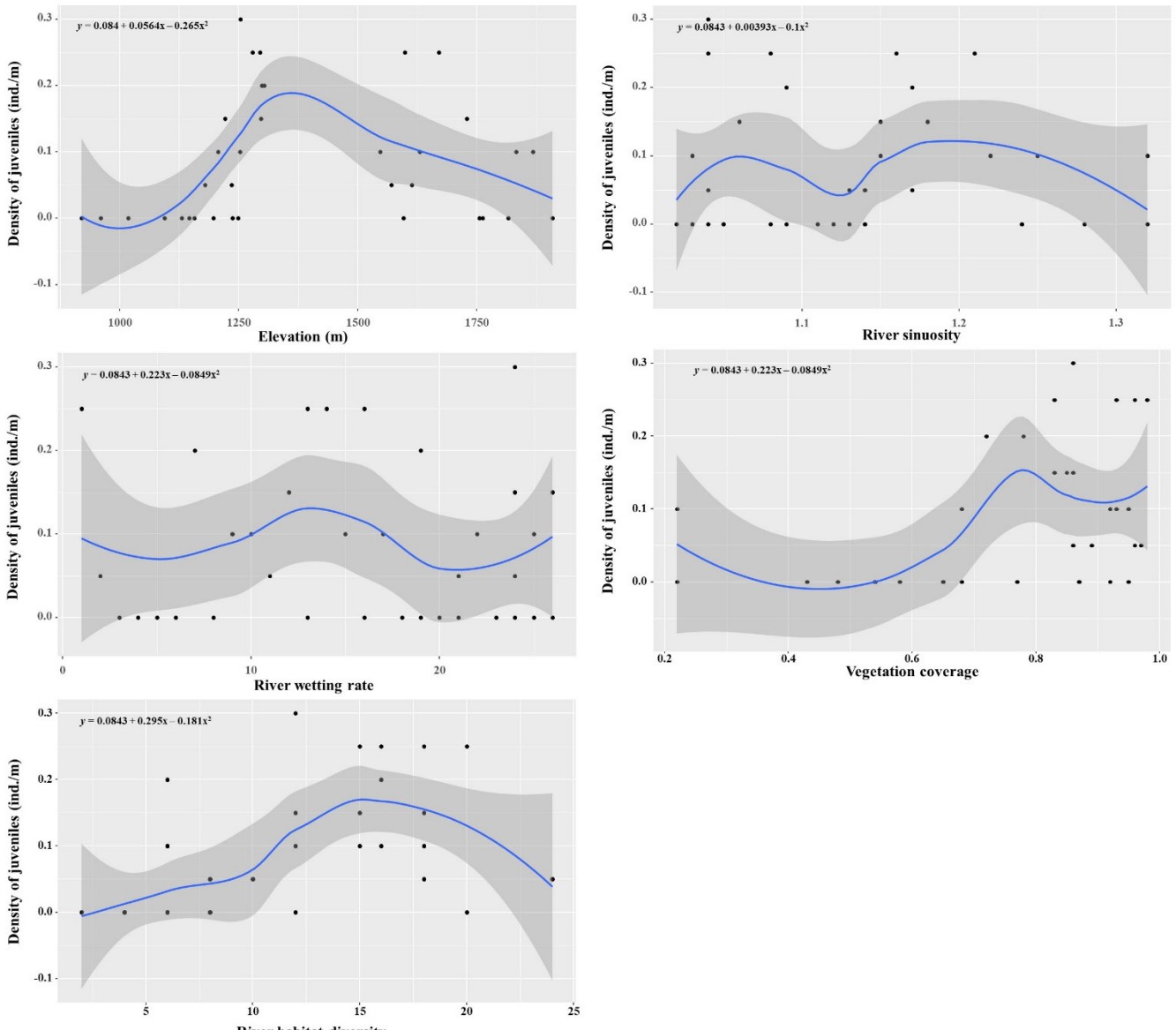

**Figure 2.** Fish habitat suitability curve between the density of juveniles and environmental factors. The blue line represents the curve for the fitting relationship between response and predictor; shaded areas indicate 95% confidence interval.

### 3.3. Microhabitat Selection Preference of Juvenile H. bleekeri

To further study the microhabitat characteristics of juvenile *H. bleekeri*, 50 marked distribution sites were selected for analysis (Table 4). The result showed that the water depth of the juvenile *H. bleekeri* habitat ranged from 0.18 to 1.40 m, with an average of 0.65 ± 0.33 m. The flow velocity ranged from 0.12 to 0.99 m/s, with an average of 0.50 ± 0.24 m/s. The illuminance ranged from 722 to 9886 Lux, with an average of 4604 ± 2670 Lux. The offshore distance varied from 0.51 to 16.65 m, with an average of 7.66 ± 4.25 m. ANOVA on the abundances of juvenile *H. bleekeri* in distinct microhabitat types showed significant differences in the preferences for water velocity, depth, and offshore distance, but not illuminance, by individuals of different size classes.

**Table 4.** Microhabitat variables for different sizes of juvenile fish.

| Individual Size | Water Depth (m) | Average | Flow Velocity (m/s) | Average | Illuminance (Lux) | Average | Offshore Distance (m) | Average |
|---|---|---|---|---|---|---|---|---|
| 5–10 cm | 0.29 | | 0.25 | | 7144 | | 4.25 | |
| | 0.85 | | 0.28 | | 9608 | | 8.6 | |
| | 0.48 | | 0.14 | | 8054 | | 2.77 | |
| | 0.6 | | 0.42 | | 6852 | | 6.43 | |
| | 0.36 | | 0.37 | | 4559 | | 5.16 | |
| | 0.43 | | 0.41 | | 2775 | | 8.71 | |
| | 0.52 | | 0.27 | | 9768 | | 2.86 | |
| | 0.31 | | 0.22 | | 4972 | | 5.05 | |
| | 0.51 | | 0.37 | | 1602 | | 5.92 | |
| | 0.89 | | 0.49 | | 7053 | | 0.51 | |
| | 0.34 | | 0.6 | | 8030 | | 5.05 | |
| | 0.52 | | 0.45 | | 6645 | | 9.57 | |
| | 0.66 | | 0.22 | | 4231 | | 5.08 | |
| | 0.78 | $0.50 \pm 0.22$ [a] | 0.44 | $0.32 \pm 0.13$ [a] | 1721 | $5313 \pm 2836$ | 3.1 | $5.58 \pm 2.50$ [a] |
| | 0.23 | | 0.53 | | 1337 | | 8.04 | |
| | 0.78 | | 0.43 | | 9886 | | 9.88 | |
| | 0.33 | | 0.27 | | 9662 | | 4.9 | |
| | 0.47 | | 0.49 | | 2767 | | 8.29 | |
| | 0.47 | | 0.34 | | 7747 | | 2.59 | |
| | 0.71 | | 0.33 | | 1287 | | 6.55 | |
| | 0.81 | | 0.27 | | 2760 | | 5.07 | |
| | 0.83 | | 0.14 | | 4245 | | 8.51 | |
| | 0.22 | | 0.23 | | 6036 | | 5.17 | |
| | 0.19 | | 0.33 | | 2863 | | 8.47 | |
| | 0.2 | | 0.2 | | 4430 | | 1.4 | |
| | 0.18 | | 0.16 | | 6365 | | 5.73 | |
| | 0.65 | | 0.12 | | 1052 | | 3.03 | |
| 11–15 cm | 0.4 | | 0.64 | | 5495 | | 4.83 | |
| | 1.4 | | 0.69 | | 6624 | | 4.36 | |
| | 0.53 | | 0.75 | | 891 | | 4.76 | |
| | 0.44 | | 0.64 | | 2940 | | 16.3 | |
| | 1.29 | | 0.65 | | 1877 | | 9.02 | |
| | 1 | | 0.62 | | 3067 | | 2.68 | |
| | 0.6 | $0.73 \pm 0.34$ [ab] | 0.7 | $0.63 \pm 0.12$ [b] | 722 | $3966 \pm 2506$ | 15.47 | $9.17 \pm 4.11$ [b] |
| | 0.68 | | 0.8 | | 966 | | 9.75 | |
| | 0.39 | | 0.76 | | 2158 | | 10.38 | |
| | 1.08 | | 0.67 | | 5639 | | 7.82 | |
| | 0.46 | | 0.53 | | 7635 | | 9.67 | |
| | 0.74 | | 0.34 | | 5842 | | 13.61 | |
| | 0.43 | | 0.44 | | 7703 | | 10.5 | |
| 16–20 cm | 1.09 | | 0.64 | | 4739 | | 10.31 | |
| | 0.83 | | 0.42 | | 1329 | | 1.17 | |
| | 1.27 | | 0.83 | | 3954 | | 16.65 | |
| | 0.44 | | 0.93 | | 1262 | | 6.92 | |
| | 0.52 | | 0.84 | | 3620 | | 13.7 | |
| | 1.3 | $0.95 \pm 0.30$ [b] | 0.99 | $0.81 \pm 0.16$ [c] | 2164 | $3517 \pm 1641$ | 15.06 | $11.32 \pm 4.86$ [b] |
| | 1.21 | | 0.97 | | 2644 | | 5.9 | |
| | 1 | | 0.9 | | 5900 | | 13.4 | |
| | 0.66 | | 0.79 | | 3387 | | 13.75 | |
| | 1.19 | | 0.82 | | 6178 | | 16.33 | |
| Total average | $0.65 \pm 0.33$ | | $0.50 \pm 0.24$ | | $4604 \pm 2670$ | | $7.66 \pm 4.25$ | |

Some data were presented as mean $\pm$ SD. [a,b,c] Mean values with unlike letters were significantly different ($p < 0.05$).

The analysis of substrate selection showed that the proportion of 5–10 cm juveniles choosing gravel was the highest, accounting for 40.74% of all sites. Further, 11–15 cm juveniles had similar selection ratios for four rocky substrates but were not found in the bedrock. There was a clear preference for pebbles in individuals sized 16–20 cm, accounting for 50% (Table 5). These results showed that juvenile *H. bleekeri* were more inclined to choose rocky substrates, and fishes of different sizes preferred different types of rocky substrates.

**Table 5.** Substrate selection percentage for different sizes of juvenile fish.

| Individual Size | Substrate Type | | | | |
| --- | --- | --- | --- | --- | --- |
| | Gr | Pe | Co | Bo | Br |
| 5–10 cm | 40.74% | 18.52% | 18.52% | 14.81% | 7.41% |
| 11–15 cm | 23.08% | 23.08% | 30.77% | 23.08% | |
| 16–20 cm | 10.00% | 50.00% | 20.00% | 10.00% | 10.00% |

## 4. Discussion

*H. bleekeri* has the lowest latitude distribution among the five members of the *Hucho* genus [25]. It has been inferred that *H. bleekeri* migrated from the north when the climate cooled during the Quaternary glacial period. After the end of the glacial period, they remained in rivers at higher elevations with lower water temperatures and differentiated into the present independent species in the Dadu River system [26]. When the original distribution area is narrow, the habitat extension available to juvenile fish decreases significantly because of changes in the environment, reductions in food resources, and overfishing [27]. Changes in the distribution area of *H. bleekeri* are the most significant among the fish species [28]. Based on the survey and resource evaluation results of this study, the *H. bleekeri* population in the Taibai River is the largest discovered over the past 30 years in China [15]. High forest coverage and abundant resources provide the necessary ecological conditions for *H. bleekeri*, and the reduction in human interference following the designation of special jurisdictional areas in the upper reaches of the Taibai River may be the main reason for preserving the habitat and population of this species. The present research indicates that *H. bleekeri* have specialized requirements regarding environmental conditions, especially at intermediate and local scales.

Correlation analysis confirmed that environmental factors affect habitat selection by juvenile *H. bleekeri*, including elevation, river sinuosity, wetting rate, vegetation coverage, and habitat diversity, which strongly correlate among them. The fish habitat suitability curve is the most important method for evaluating fish habitat suitability, reflecting the contribution of key factors to the habitat [29]. Our results also confirmed that the selected environmental factors correlated with the density of juvenile *H. bleekeri*. Elevation reflects habitat environmental attributes at a macrogeographic scale. The suitable elevation range calculated in this study suggests that *H. bleekeri* may adapt to higher-elevation rivers, which is consistent with previous research [30]. However, there was a gap at the elevation between 1300 and 1600 m, probably because of the relatively low vegetation coverage in this region. The river sinuosity can create a rich and diverse habitat of bays, marshes, and shoals [31], and suitable river habitat diversity may provide adequate food sources for fish and play a role in the regulation of the water quality, which may in turn affect the distribution of fish [32]. Our results confirmed that river sinuosity and habitat diversity are important factors affecting the density of juvenile *H. bleekeri*. It has been reported that higher vegetation coverage can reduce the rise in the water temperature caused by sunlight, which prompts *Oncorhynchus masou formosanus* to hide in the summer [33]. In addition, vegetation on both sides of the stream maintains the relative humidity and provides a habitat for a number of terrestrial insects [34]. Thus, natural vegetation cover may be an essential factor influencing the density of juvenile *H. bleekeri*.

Microhabitat selection is an important factor affecting the coexistence of species with similar habitat requirements and is specific for different salmonid species and also between life stages within the same species [35]. The type of substrate is an important factor affecting the growth of fish [36,37]. A riverbed with a wide range of sediment particle sizes provides various associated prey organisms as well as numerous refuges for the fish [38]. Various salmonid species exhibit different preferences for different substrate types. *Oncorhynchus masou formosanus* is accustomed to inhabiting water bottoms with gravel [39]. *Brachymystax lenok* from the Nakdong River prefers to inhabit water bottoms with bedrock or drift [40]. However, substrate preferences vary depending on the river environment. The present study shows that juvenile *H. bleekeri* inhabits various substrate types, including gravel, pebbles, cobbles, boulders, and bedrock, which indicates that complex substrate environments may be more suitable habitats for juvenile *H. bleekeri*. The preference of juvenile fish of different sizes for gravel or pebble substrate may be related to their size-related access to prey.

The results of several studies showed that the fish distribution may be influenced by factors such as the water velocity, depth, illuminance, and offshore distance [29,39]. In the present study, the water velocity, depth, and offshore distance significantly differed among juvenile fish of different size. To some extent, changes in the water depth represent changes in the offshore distance. A greater offshore distance or deeper water may provide favorable conditions for larger juvenile *H. bleekeri* to manoeuvre and to hide. In contrast, low water velocity may be beneficial for smaller juvenile individuals by reducing their energy consumption. Illuminance is an essential environmental factor affecting the distribution of fish. It has been reported that high light intensity may lead to *Pelteobaggrus vachelli*-aggregating behavior [41]. At low light intensity, *P. vachelli* exhibits swimming and cluster dispersion behaviors. This phenomenon was observed in *Procypris rabaudi* [42]. In the present study, no significant differences were observed between juvenile *H. bleekeri* with various sizes and illuminance levels. Therefore, whether juvenile *H. bleekeri* exhibits aggregating behavior at various light intensities requires further confirmation.

## 5. Conclusions

In this study, the distribution and habitat requirements of juvenile *H. bleekeri* in the Taibai River were determined based on a literature review and field investigation. Elevation, river sinuosity, and vegetation coverage are the key environmental factors affecting the distribution of juvenile fish. Juvenile fish with different sizes make different choices in terms of the substrate type, flow velocity, water depth, and offshore distance. Smaller juvenile fish tend to choose nearshore habitats with gravel substrate, lower flow velocities, and shallower water depths. As this study was only carried out at the end of the feeding period of *H. bleekeri*, the results could not fully describe the relationship between juvenile fish and environmental factors. Climate change may have a more profound impact on the distribution of this cold water fish species; thus, further research during other seasons is still necessary. Regardless of this limitation, the results of our study indicate that juvenile *H. bleekeri* display strong association with specific habitat types, which might be the main factor limiting their distribution. Therefore, strict protection of their historical and potential distribution areas is of great practical significance for the conservation and restoration of this species.

**Author Contributions:** Conceptualization, methodology, software, investigation, resources and writing—original draft preparation, J.W. (Jinming Wu), H.D., W.X., H.Y. and W.Q.; writing—review, editing, and visualization, X.L. and J.W. (Jinping Wu); supervision, project administration, funding acquisition, J.W. (Jinming Wu) and X.L. All authors have read and agreed to the published version of the manuscript.

**Funding:** This research was funded by the National Key R&D Program of China (grant numbers 2021YFD1200304 and 2021YFD1200305) and National Natural Science Foundation of China (grant number 31602165).

**Institutional Review Board Statement:** The animal study protocol was approved by the Animal Experimental Ethical Committee of the Laboratory Animal Center, Yangtze River Fisheries Research Institute, Chinese Academy of Fishery Sciences, China. Approval code: yfilxq03.

**Data Availability Statement:** All data generated during this study are included in this published article.

**Conflicts of Interest:** The authors declare no conflict of interest.

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
