# Peer review of "Correlations between Environmental Factors and the Distribution of Juvenile Hucho bleekeri in the Taibai River, Shaanxi, China"

_fishes, doi:10.3390/fishes8070379_

Round 1
Reviewer 1 Report
Please see the attached review comments

Minor editing of English language necessary
Reviewer 2 Report
This manuscript describes a study of environmental variables where juvenile Sichuan taiman fishes occurred in 2017-18. The methods appear to be sound. The analyses need to be clarified with citations so a reader can understand clearly. I have made multiple suggestions for clarity, mostly minor English language issues.
Specific suggestions (in addition to edits on the PDF):
Lines 27-28: human wading and illegal fishing are not the main risk factors in other locations. Primarily hydrologic alteration and pollution in the US.
Line 59: Please use a different word than “bait”. Perhaps food resources. Replace bait in all locations please. Please clarify “pest-avoidance”. Do you mean avoid parasites? Or avoid predation?
Line 74: reword: “upstream and downstream in the river, respectively.”
Line 87: reword: “returned to the river”; describe that density of fish was calculated as individuals per m stream distance.
Line 90: replace “indicators” with “variables”; and elsewhere in the manuscript.
Line 110: what fitness curve? Is this described later – please state this. Is this the suitability curve in the Results? Use the same term please.
Line 117: replace “traces” with “movements” or “locations”
Lines 122-4: was substrate defined visually, or did you measure sizes?
Line 143: replace “singular” with “simple”
Line 146: use past tense here and elsewhere
Table 3: state in the legend what the asterisks represent please.
Line 171: The methods or results need to describe how a suitability curve represents correlations.
Line 186: past tense. This sentence does not describe the data in the figure. Gravel substrates were most common.
Line 209-10: does this refer to fish species in this river?
Line 211: delete “group”
Line 216: replace “higher” with “specialized”; replace “microscopic” with “local”
In general the English is excellent. I made editorial suggestions in locations described in my review. One major issue is the manuscript needs to be past tense.
Reviewer 3 Report
This is valuable work on endangered freshwater salmonid. You can see my Line-by-Line review at the attached file.

Round 2
Reviewer 1 Report
The authors have satisfactorily addressed the comments and suggestions provided during the first round of review.
Minor editing of English required
Author Response
The new revision has been English edited as suggested.
Reviewer 2 Report
Second review of “Correlations between environmental factors and the distribution of juvenile Hucho bleekeri in the Taibai River”
Authors made substantial revisions. This manuscript describes a study of environmental variables where juvenile Sichuan taiman fishes occurred in 2017-18. I made additional editorial suggestions or recommended authors use my original suggestions.
Specific suggestions:
Lines 27-28: Thanks for your modifications. Water engineering does not include pollution and multiple other human activities that harm aquatic ecosystems. I suggest replacing “water engineering” with “multiple human impacts”.
Line 61: Feed is not a good word here. This implies something more like feed for cows or horses. I recommend food resources.
Line 63: delete “certain”
Lines 131-3: was substrate defined visually, or did you measure sizes? This needs to be clarified please. Substrate measured by tape does not explain the method – and it needs to be in the text.
Line 176-7: The methods or results need to describe how a suitability curve represents correlations. This information needs to be in the Methods or in the Results sections please. On lines 225-7 you describe the fish habitat suitability curve. This does not explain how to interpret this information.
Line 209-10: does this refer to fish species in this river?
Line 216: replace “nationwide” with a country name please
Many issues are still present in the manuscript.
Reviewer 3 Report
Authors make some improvements and paper can be published.
Author Response
Thank you for your comments and approval.